# Healthcare-associated infection and its determinants in Ethiopia: A systematic review and meta-analysis

**Abebaw Yeshambel Alemu**[1]*, **Aklilu Endalamaw**[2], **Demeke Mesfin Belay**[1], **Demewoz Kefale Mekonen**[1], **Biniam Minuye Birhan**[1], **Wubet Alebachew Bayih**[1]

**1** Department of Paediatrics and Neonatal Health Nursing, College of Health Sciences, Debre Tabor University, Debre Tabor, Ethiopia, **2** Department of Paediatrics and Child Health Nursing, School of Health Sciences, College of Medicine and Health Sciences, Bahir Dar University, Bahir Dar, Ethiopia

* yeshambelabebaw@gmail.com

**Data Availability Statement:** All relevant data are within the manuscript and its Supporting Information files.

## Abstract

### Background

Healthcare-associated infection is a global threat in healthcare which increases the emergence of multiple drug-resistant microbial infections. Hence, continuous surveillance data is required before or after patient discharge from health institutions though such data is scarce in developing countries. Similarly, ongoing infection surveillance data are not available in Ethiopia. However, various primary studies conducted in the country showed different magnitude and determinants of healthcare-associated infection from 1983 to 2017. Therefore, this systematic review and meta-analysis aimed to estimate the national pooled prevalence and determinants of healthcare-associated infection in Ethiopia.

### Methods

We searched PubMed, Science Direct, Google Scholar, and grey literature deposited at Addis Ababa University online repository. The quality of studies was checked using Joanna Brigg's Institute quality assessment scale. Then, the funnel plot and Egger's regression test were used to assess publication bias. The pooled prevalence of healthcare-associated infection was estimated using a weighted-inverse random-effects model meta-analysis. Finally, the subgroup analysis was done to resolve the cause of statistical heterogeneity.

### Results

A total of 19 studies that satisfy the quality assessment criteria were considered in the final meta-analysis. The pooled prevalence of healthcare-associated infection in Ethiopia as estimated from 18 studies was 16.96% (95% CI: 14.10%-19.82%). In the subgroup analysis, the highest prevalence of healthcare-associated infection was in the intensive care unit 25.8% (95% CI: 3.55%-40.06%) followed by pediatrics ward 24.16% (95% CI: 12.76%-35.57%), surgical ward 23.78% (95% CI: 18.87%-29.69%) and obstetrics ward 22.25% (95% CI: 19.71%-24.80%). The pooled effect of two or more studies in this meta-analysis also showed that patients who had surgical procedures (AOR = 3.37; 95% CI: 1.85–4.89)

**Funding:** The author(s) received no specific funding for this work.

**Competing interests:** The authors have declared that no competing interests exist.

**Abbreviations:** AOR, Adjusted Odds Ratio; CI, Confidence Interval; HCAI, Healthcare-Associated Infection; ICU, Intensive Care Unit; SNNPR, Sothern Nations Nationalities and Peoples Region; WHO, World Health Organization.

and underlying non-communicable disease (AOR = 2.81; 95% CI: 1.39–4.22) were at increased risk of healthcare-associated infection.

## Conclusions

The nationwide prevalence of healthcare-associated infection has remained a problem of public health importance in Ethiopia. The highest prevalence was observed in intensive care units followed by the pediatric ward, surgical ward and obstetrics ward. Thus, policymakers and program officers should give due emphasis on healthcare-associated infection preventive strategies at all levels. Essentially, the existing infection prevention and control practices in Ethiopia should be strengthened with special emphasis for patients admitted to intensive care units. Moreover, patients who had surgical procedures and underlying non-communicable diseases should be given more due attention.

## Introduction

According to the Communicable Diseases Control (CDC), healthcare-associated infection (HCAI) is defined as the acquisition of infectious agent(s) or its toxin(s) which occurs after 48 hours of hospital admission, or up to 3 days after discharge, or up to 30 days after the operation when someone was admitted for reasons other than infection [1, 2].

wGlobally, according to the World Health Organization (WHO) 2019 HCAI fact sheet report, a hundred million patients were affected each year [3]. The point prevalence of HCAI ranged from 3.5%-12% and 5.7%-19.1% in developed and Low-and Middle-Income Countries (LMICs), respectively [3, 4]. Though data is scarce, the burden of HCAI was found to be high in Sub-Saharan Africa (SSA) countries [5]. Specifically, the prevalence of HCAI was noted in Botswana (13.4%) [6], South Africa (8%) [7], and Ethiopia (13% to 22%) [8–10].

Healthcare-associated infection increases the occurrence of antimicrobial resistance [11], long-term disability [4], and mortality among individual patients [12]. The additional financial burden to the healthcare system, patients, and families due to HCAI is also significant [4]. Hence, the "*clean care the safer care*" program has been launched in 2004 with the WHO patient safety directive, which was aimed to reduce HCAI through improving hand hygiene practice at the center of achieving its aim [13]. The aforementioned infection prevention program and the WHO initiative about infection prevention and control policy recommendations have been implemented in developing countries, including Ethiopia. Despite these efforts, studies conducted at different settings of the globe revealed that admission to the surgical ward and hospital type [8], chest tube placement, prolonged hospital stays, patient on mechanical ventilation, previous hospitalization [9], pediatric patients, malnutrition, and length of staying in hospital >5days [10] were contributing factors of HCAI.

Various studies were conducted to determine the prevalence of HCAI in Ethiopia, but it showed great variation across geographical setting and variant periods. Based on this fact, there was a need for nationally representative data on HCAI in the country. Moreover, the pooled effect sizes of the determinants of HCAI weren't explored nationwide. Consequently, this systematic review and meta-analysis was aimed to address the following research questions: (1) what is the national pooled prevalence of HCAI in Ethiopia; and (2) what are the determinants of HCAI in the country?

## Materials and methods

### Reporting

The study results were reported based on the Preferred Reporting Items for Systematic Review and Meta-analysis statement (PRISMA) guideline [14] (S1 File). The protocol was registered on the PROSPERO database with a registration number (CRD42020166761), and available on https://www.crd.york.ac.uk/prospero/display_record.php?ID=CRD42020166761.

### Inclusion and exclusion criteria

We included cross-sectional, case-control and cohort studies, but case-control studies weren't used to estimate the pooled prevalence of HCAI. These studies were included when the prevalence, incidence, and/or at least one determinant was reported. All studies published in the English language were considered. There was no restriction of the study period, age group, and study setting. All citations without abstract and/or full-text, anonymous reports, editorials, and qualitative studies were excluded.

### Search strategy and information source

PubMed, Science Direct, Google Scholar, and grey literature deposited at Addis Ababa University online repository were searched. The core search terms and phrases were "prevalence", "incidence", "epidemiology", "proportion", "magnitude", "burden", "associated factors", "risk factors", "predictors", "determinants", "healthcare-associated infections", "healthcare-acquired infections", and "nosocomial infections", "hospital acquired infections" and "Ethiopia". The search strategies were developed using different Boolean operators. Notably, to fit the advanced PubMed database, the following search strategy was applied: [(prevalence) OR incidence[MeSH Terms]) OR epidemiology[MeSH Terms]) OR proportion[MeSH Terms]) OR magnitude[MeSH Terms]) OR burden[MeSH Terms]) AND associated factors) OR risk factors[MeSH Terms]) OR predictors[MeSH Terms]) OR determinants[MeSH Terms]) AND healthcare-associated infections) OR healthcare-acquired infections[MeSH Terms]) OR nosocomial infections[MeSH Terms]) OR hospital acquired infections[MeSH Terms]) AND (Ethiopia)]. Then, we retrieved 611 articles using this PubMed searching strategy.

### Study selection

Duplicate studies were removed using Endnote version 8 (Thomson Reuters, London) reference manager software. The two independent reviewers (AYA and WAB) screened the titles and abstracts. The disagreements were handled based on established article selection criteria. Then, two independent authors (AE and BMB) conducted the abstracts and full-texts review.

### Quality assessment

The two independent authors (DKM and DMB) appraised the quality of the studies. The Joanna Briggs Institute (JBI) quality appraisal checklist was used [15]. The disagreement was resolved by the involvement of a third reviewer (AE). To appraise cohort studies, the following items were used: (i) similarity of groups;(ii) similarity of exposure measurement;(iii) validity and reliability of measurement;(iv) identification of confounder;(v) strategies to deal with confounder;(vi) appropriateness of groups/participants at the start of the study;(vii) validity and reliability of outcome measured;(viii) sufficiency of follow-up time;(ix) completeness of follow-up or descriptions of reason to loss to follow-up;(x) strategies to address incomplete follow-up; and (xi) appropriateness of statistical analysis. The items used to appraise case-control studies were: (i) comparable groups;(ii) appropriateness of cases and controls;(iii) criteria to

identify cases and controls;(iv) standard measurement of exposure;(v) similarity in the measurement of exposure for cases and controls; (vi) handling of confounders;(vii) strategies to handle confounder;(viii) standard assessment of outcome;(ix) appropriateness of duration for exposure; and (x) appropriateness of statistical analysis. Cross-sectional studies were appraised based on (i) inclusion criteria;(ii) description of study subject and setting;(iii) valid and reliable measurement of exposure; (iv) the objective and standard criteria used;(v) identification of confounder;(vi) the strategies to handle confounder; (vii) outcome measurement; and (viii) appropriate statistical analysis. All the studies which got 50% and above on the quality assessment scale were considered as low risk.

## Data extraction

Two independent reviewers (AYA and AE) extracted data using a structured data extraction form. Whenever variations of extracted data were observed, the phases were repeated. If discrepancies between data extractors continued, the third reviewer (WAB) was involved. The name of the first author and year, study region, study design, target population, diagnostic methods, sample size, the prevalence of HCAI, and adjusted odds ratio (AOR) of associated factors were collected.

## Outcome measurement

HCAI was considered when reported as infection(s) acquired while receiving medical care based on culture-confirmation [10, 16–18], or clinical and laboratory methods [8, 9, 19–31].

## Statistical analysis

Publication bias was checked visually by the funnel plot, and objectively using Egger's regression test [32]. Heterogeneity of studies was quantified using the I-squared statistic, in which 25%, 50%, and 75% represented low, moderate, and high heterogeneity, respectively [33]. Pooled analysis was conducted using a weighted-inverse variance random-effects model [34]. The subgroup analysis was done by region, study design, diagnostic method, sample size and ward type. Sensitivity analysis was employed to see the effect of a single study on the overall estimation. Besides, the time-trend analysis was conducted to check the variation through time. STATA version 11 statistical software was used for meta-analysis.

## Ethics approval and consent to participate

Not applicable because no primary data were collected from patients.

## Results

### Literature search

The search strategy retrieved 611 articles from PubMed, 133 from Science Direct, 19 from Google Scholar, and 3 grey literatures from Addis Ababa University online repository. After duplicates were removed, 740 studies remained. Then, sixty studies were screened for full-text review. Finally, 19 studies were used in the systematic review and/or meta-analysis (**Fig 1**).

### Characteristics of the included studies

Six studies were found in Addis Ababa [19–21, 23, 25, 31], five studies in Amhara region [8, 16, 26, 27, 29], five studies in Oromia [9, 17, 18, 28, 30], one study both in Addis Ababa and Southern Nation Nationalities and People Region (SNNPR) [22], one each in Tigray [24] and

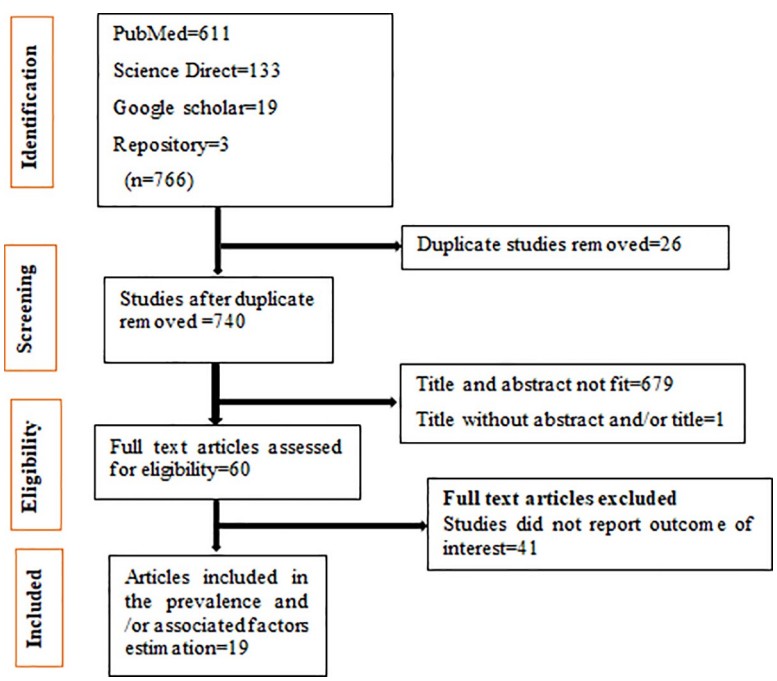

**Fig 1. The study selection process.**

SNNPR [10]. Nine studies were conducted across all age groups. Eight studies were done on the adult population and one study was on pediatric patients. Fourteen studies used clinical and laboratory methods for the diagnosis of HCAI while the remaining were culture-confirmed. Four studies were conducted using cohort study design, fourteen were cross-sectional and only one was a case-control study. Only six studies had >1000 sample size (**Table 1**).

## Quality of studies

The JBI quality appraisal criteria established for cross-sectional, case-control, and cohort studies were used. The studies included in this systematic review and meta-analysis had no considerable risk. Therefore, all the studies were considered [8–10, 16–31] (**Table 1**).

## Meta-analysis

**Publication bias.** The funnel plot showed symmetrical distribution (**Fig 2**). Egger's regression test p-value was 0.328, which indicated the absence of publication bias.

**The prevalence of healthcare-associated infection.** *A total of 18 studies were used and 14,240 patients participated in the* prevalence estimation. The estimated overall prevalence of HCAI is presented in a forest plot (**Fig 3**). The overall prevalence of HCAI was 16.96% (95% confidence interval (CI): 14.10%-19.82%).

**Subgroup analysis.** The subgroup analyses based on study region, study design, diagnostic method, and the sample size were done. Accordingly, the prevalence of HCAI was found 27.6% in Tigray region, 18.2% diagnosed by clinical and laboratory methods, 17.83% in the cross-sectional studies, 18.15% in studies using < 1000 study samples (**Table 2**).

The prevalence of HCAI was reported across various wards too. A study conducted at Jimma University Hospital showed that the incidence of HCAI was the highest in the Intensive Care Unit (ICU) (207.55/1000 patient-days) followed by the pediatric ward (69.16/1000

Table 1. Characteristics and quality status of the studies included.

| First author year | Study region | Study design | Sample size | Prevalence | Quality status |
|---|---|---|---|---|---|
| Gedebu M. et al./1987 [19] | Addis Ababa | Cross-sectional | 2506 | 13.40 | Low risk |
| Gedebu M. et al./1988 [20] | Addis Ababa | Cross-sectional | 700 | 17.00 | Low risk |
| Habte-Gaber E. et al./1988 [21] | Addis Ababa | Cohort | 1006 | 16.40 | Low risk |
| Berhe N. et al./2001 [22] | Addis Ababa and SNNPR | Cohort | 247 | 5.90 | Low risk |
| Endalfer N. et al./2008 [23] | Addis Ababa | Cross-sectional | 854 | 9.00 | Low risk |
| Tesfahun Z. et al./2009 [24] | Tigray region | Cross-sectional | 246 | 27.60 | Low risk |
| Endalfer N. et al./2011 [25] | Addis Ababa | Cross-sectional | 215 | 35.80 | Low risk |
| Melaku S. et al./2012 [26] | Amhara region | Cross-sectional | 1383 | 17.80 | Low risk |
| Melaku S. eta al/2012 [27] | Amhara region | Cross-sectional | 1254 | 9.40 | Low risk |
| Mulu W. et al./2013 [16] | Amhara region | Cross-sectional | 294 | 10.90 | Low risk |
| Sahile T. eta al/2016 [28] | Oromia region | Cross-sectional | 500 | 35 | Low risk |
| Yallew WW. et al./2016 [8] | Amhara region | Cross-sectional | 908 | 14.90 | Low risk |
| Tolera M. et al./2018 [18] | Oromia region | Cross-sectional | 394 | 6.90 | Low risk |
| Gashaw M. et al./2018 [17] | Oromia region | Cross-sectional | 1015 | 11.60 | Low risk |
| Ali S. et al./2018 [9] | Oromia region | Cohort | 1069 | 19.40 | Low risk |
| Alemayehu T. et al./2019 [10] | SNNPR | Cross-sectional | 939 | 21.40 | Low risk |
| Gebremeskel S. et al./2018 [31] | Addis Ababa | Cross-sectional | 410 | 19.80 | Low risk |
| Yallew WW. et al./2017 [29] | Amhara region | Case-control | 545 | | Low risk |
| Zewdu et al./2017 [30] | Oromia region | Cohort | 300 | 14.00 | Low risk |

**Note:** SNNPR: Southern Nations Nationalities and Peoples Region; Low risk: a study scored > 50% in the JBI quality assessment scale.

patient-days), and surgical ward (28.87/1000 patient-days) [9]. In two studies, Yallew WW. et al. [8] and Ali S. et al. [9], HCAI was the lowest in the ophthalmology ward.

Besides, in this meta-analysis, HCAI was estimated in different wards based on the pooled effect of two or more studies. As estimated from the effect of two studies [25, 30], the prevalence of HCAI was the highest in ICU (25.8%) followed by pediatrics (24.16%) [8, 10, 31], surgical (23.78%) [8, 19, 21, 23, 25, 26, 28, 30, 31] and obstetrics ward (22.25%) [19, 26] (**Fig 4**).

**Sensitivity analysis.** The studies of Endalafer N. et al. [25] and Sahile T. et al. [28] had shown an impact on the overall estimate of HCAI (**Fig 5**).

**Time-trend analysis.** The time-trend analysis showed that the prevalence of HCAI was increased from 13.4% in 1983 to 19.8% in 2017. However, the pooled prevalence was not increasing significantly from year to year (p-value: 0.620) (**Fig 6**).

**Determinants of healthcare-associated infection.** In this systematic review and meta-analysis, HCAI in the Ethiopian context is associated with socio-demographic, patient health condition, and healthcare-related risk factors. Thus, based on the report of a single study, the age range of the patient 18–30 years was found to be protective (AOR = 0.54; 95% CI: 0.22–0.85) [9] (**Table 3**). On the contrary, based on the reports of individual studies included, HCAI had shown a positive association with the following healthcare-related factors: taking prophylaxis (AOR = 1.76; 95% CI: 1.21–2.3) [27]; admission to the surgical ward (AOR = 2.86; 95 CI: 1.33–4.38) [8]; admission at Felege Hiwot Referal Hospital (FHRH) (AOR = 1.99; 95% CI: 1.2–2.77) [8] and chest tube insertion (AOR = 4.14; 95% CI: 1.57–6.71) [9].

Moreover, in this meta-analysis, the determinants of HCAI were identified based on the pooled effect of two or more studies. Hence, as estimated from the pooled effect of two studies [25, 29], HCAI was 3.37 times (AOR = 3.37; 95% CI: 1.85–4.89) more likely among patients who had the surgical procedure as compared to no surgical procedure. Similarly, based on the pooled effect of three studies [9, 16, 27], patients who had underlying non-communicable

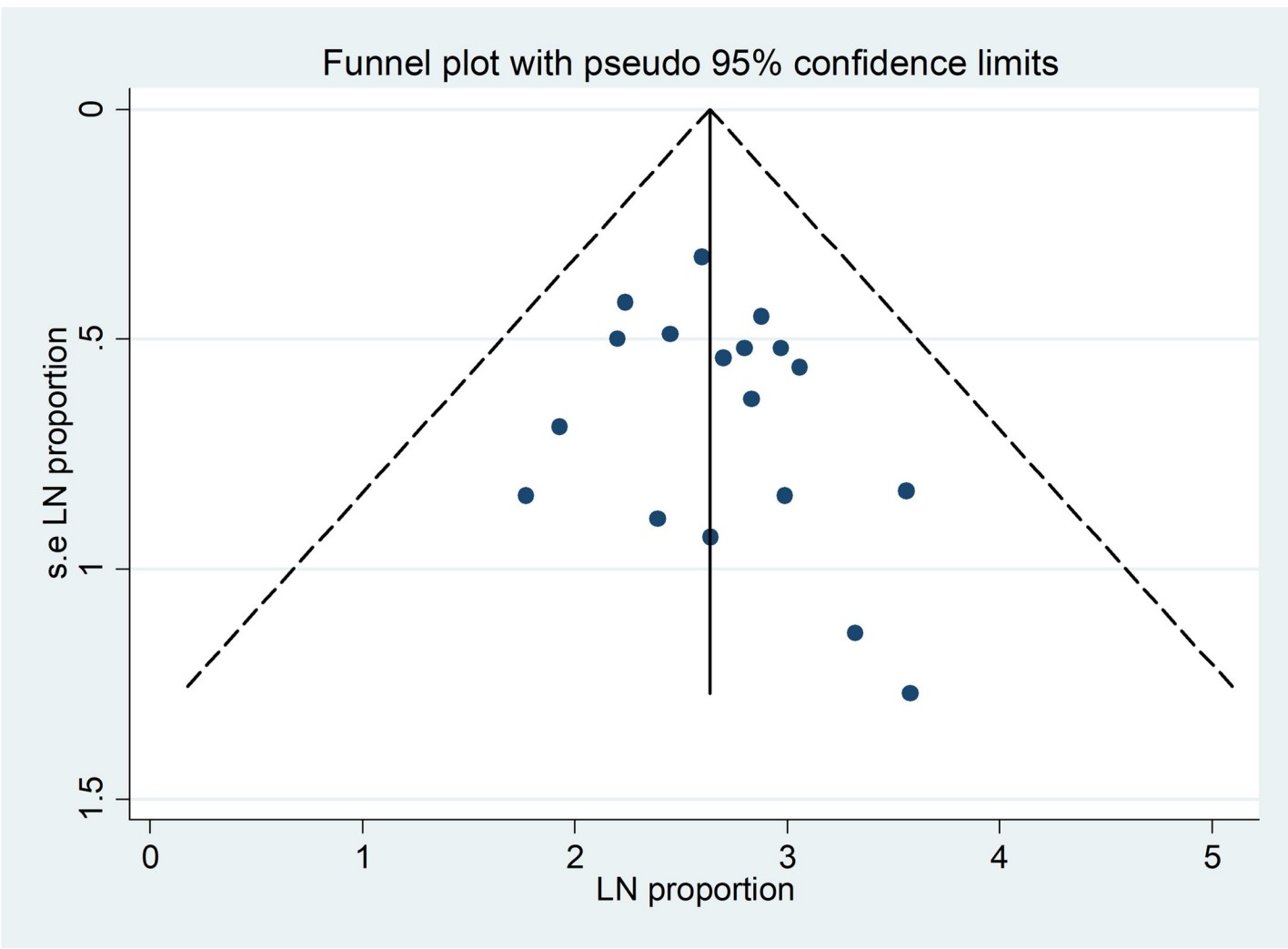

**Fig 2.** Funnel plot for publication bias, LN of proportion (X-axis) with its standard error of LN of proportion (Y-axis).

disease were 2.81 times more likely to have HCAI as compared to those without the underlying disease (**Table 3**).

## Discussion

In this systematic review and meta-analysis, the pooled prevalence of HCAI was 16.96% in Ethiopia. The authors also found that surgical procedures and underlying non-communicable diseases were identified as determinants of HCAI.

From the study, the national pooled prevalence of HCAI in Ethiopia was 16.96% (95% CI: 14.10%-19.82%). The result was higher than studies conducted in China (3.12%) [35], Morocco (10.3%) [36], Botswana (13.54%) [6], and South Africa (7.67%) [7]. The possible reasons for high prevalence in this study might be very low hand hygiene practice by physicians and resource constraints [37], low adherence to infection prevention practice [38], low level of job satisfaction [39], morally distressed nurses [40], and low implementation of the nursing process [41] in our settings, and also less attention given to HCAI.

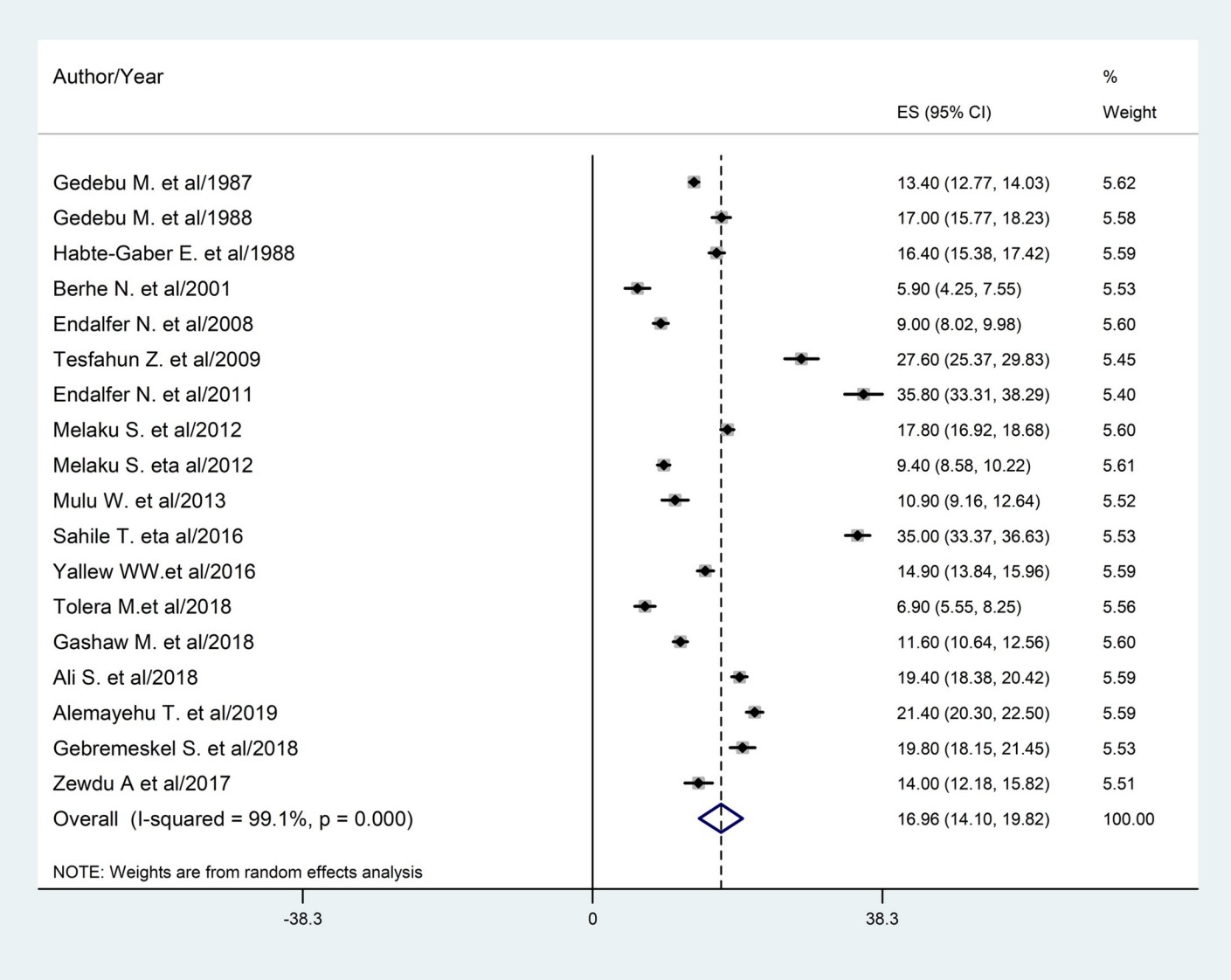

**Fig 3. Forest plot of the pooled prevalence (ES) of HCAI with corresponding 95% CIs.**

Regarding hand hygiene, only 7% of the physicians working at two University hospitals in the capital of Ethiopia, Addis Ababa, performed hand hygiene before patient contact [37]. As a result, the acquisition of HCAI from these healthcare professionals might be high. Evidence also showed that 35% of the nurses in southwest Ethiopia were non-adherent to infection prevention practice [38], thereby contributing to high HCAI in Ethiopia. Besides, nearly 68% of the health professionals were less satisfied with their work in one of the regions in the country [39]. Hence, the nosocomial infection becomes inevitably high because these less satisfied health professionals are less likely to deliver quality healthcare. Additionally, about 84% of the nurses in the northwestern part of the country [40] were morally distressed, thus causing HCAI as morally less prepared nurses were unable to deliver quality nursing care.

Resource constraints could also increase HCAI in the country because lack of hand hygiene agents and sinks were reported as hindering factors of infection prevention practice in Addis Ababa, Ethiopia [37]. Implementation of the nursing process was below half (49%) in the

**Table 2. The pooled prevalence of HCAI, 95% CI, and heterogeneity estimate with a p-value for the subgroup analysis, by region, study design, sample size, and diagnostic method.**

| Variables | Characteristics | Pooled prevalence (95% CI) | I² |
|---|---|---|---|
| Region | Addis Ababa | 18.44% (14.02–22.86) | 99% |
| | Oromia | 17.37% (9.2–25.56) | 99.5% |
| | Amhara | 13.27% (9.00–17.52) | 98.5% |
| | Tigray | 27.6% (25.37–29.83) | - |
| | SNNPR | 21.4% (20.3–22.5) | - |
| | Addis Ababa & SNNPR | 5.9% (4.25–7.55) | - |
| Study design | Cross-sectional | 17.83% (14.39–21.27) | 99.3% |
| | Cohort | 13.96% (8.78–19.14) | 98.4% |
| Diagnostic method | Clinical and laboratory | 18.2% (14.85–21.51) | 99.2% |
| | Culture-confirmed | 12.71% (6.4–19.02) | 99% |
| Sample size | <1000 | 18.15% (13.28–23.03) | 99.3% |
| | ≥1000 | 14.66% (11.72–17.59) | 98.6% |

**Note:** SNNPR: Southern Nations Nationalities and Peoples Region; I²: reported for the pooled effect of two or more studies.

northwest part of the country [41], so nursing intervention would not be planed for patients at risk of nosocomial infection. Furthermore, healthcare providers, patients, and/or families are more curious about the primary reason for admission or healthcare visits, so less attention is given to HCAI.

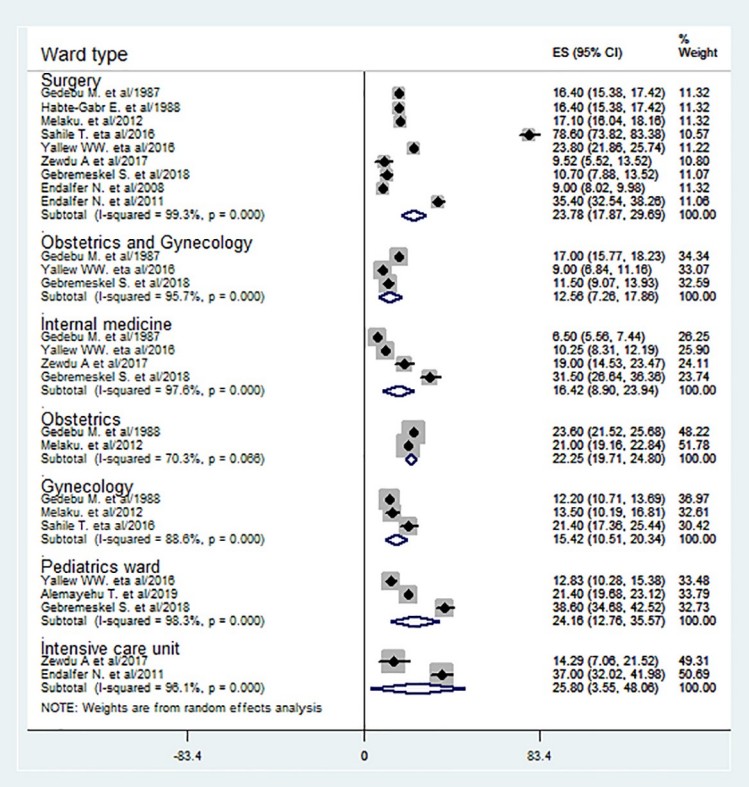

**Fig 4. The prevalence of HCAI is based on the subgroup analysis by ward type with corresponding 95% CIs.**

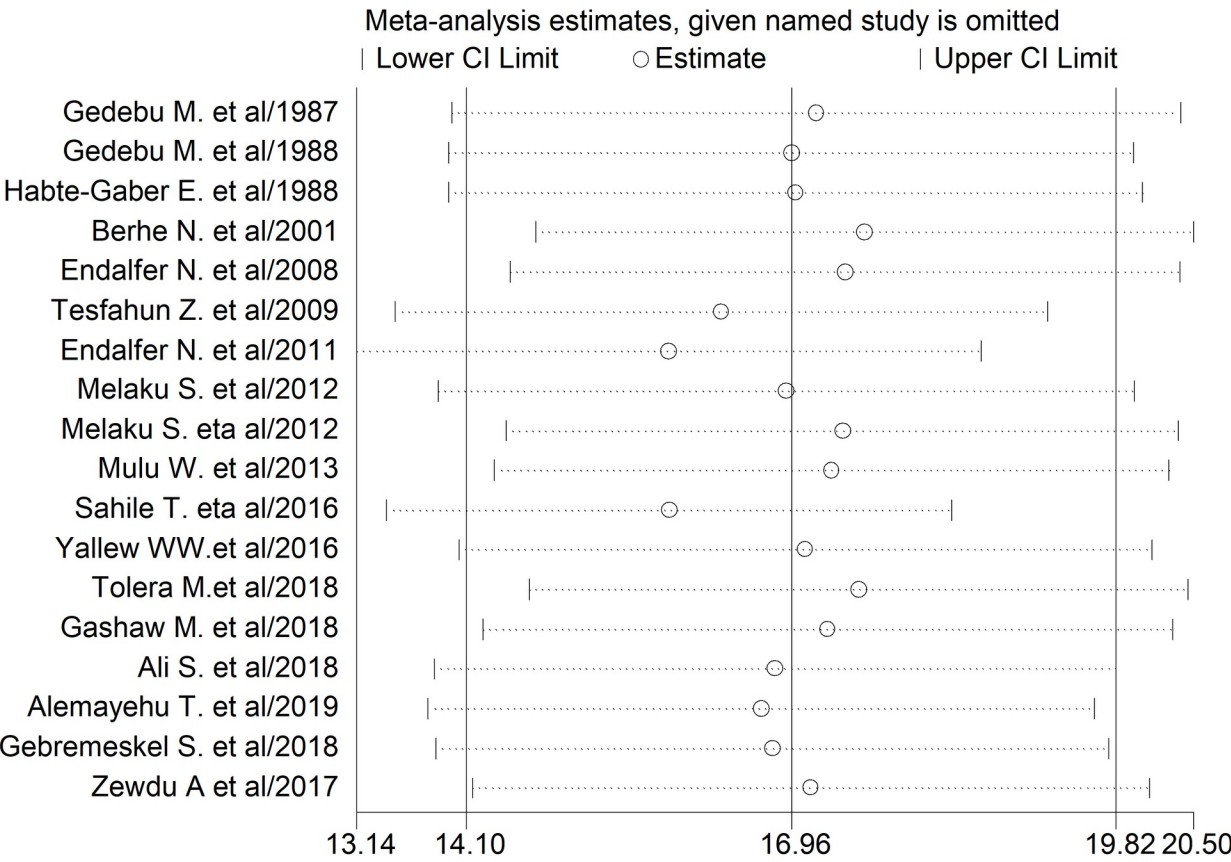

Fig 5. The sensitivity analysis showed the pooled prevalence when the studies were omitted step by step.

From the subgroup analysis, HCAI was found the highest in ICU (26%). This finding is consistent with studies conducted in China [42], India [43], and Morocco [36]. The reasons for high HCAI in ICU may be due to the highest incidence of HCAI, the severity of the disease, and highly invasive procedures. The incidence of HCAI at a referral medical center in Jimma

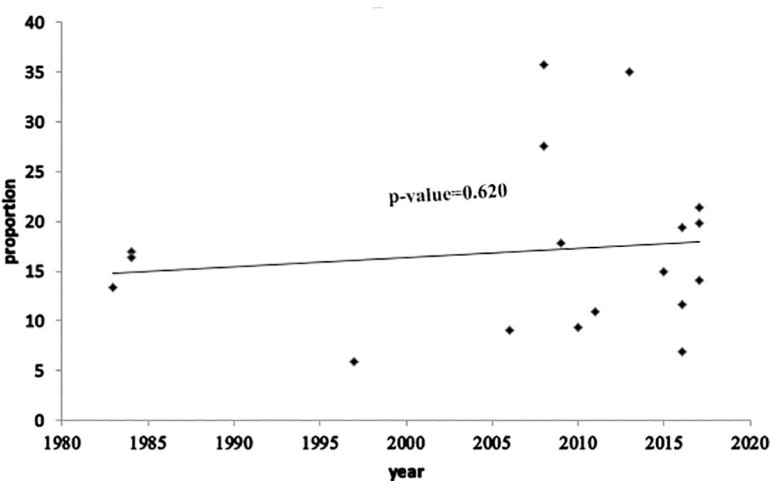

Fig 6. Time-trend analysis of the prevalence of HCAI in Ethiopia from 1983 to 2017.

**Table 3. Determinants of healthcare-associated infection in Ethiopia.**

| Determinants | Author/year | HCAI | | Effect size (95% CI) | Pooled effect size (95% CI) | $I^2$ |
|---|---|---|---|---|---|---|
| | | Yes | No | | | |
| Surgical procedure | Endalfer N. eta al/2011 [25] | 71 | 66 | 3.96 (2.82–5.09) | 3.37 (1.85–4.89) | 46.9% |
| | Yallew WW. et al./2017 [29] | 64 | 164 | 2.35 (0.35–4.34) | | |
| Take prophylaxis | Melaku S. et al./2012 [27] | 54 | 237 | 1.76 (1.21–2.3) | 1.76 (1.21–2.3) | - |
| underlying non-communicable disease | Melaku S. et al./2012 [27] | 16 | 71 | 4.3 (2.32–6.28) | 2.81 (1.39–4.22) | 54.5% |
| | Mulu W. et al./2013 [16] | 10 | 36 | 2.72 (0.42–5.01) | | |
| | Ali S. et al./2018 [9] | 44 | 135 | 2.01 (1.15–2.87) | | |
| Age ≥ 51 years | Mulu W. et al./2013 [16] | 6 | 16 | 6.38 (-10.61–23.37) | 6.38 (-10.61–23.37) | - |
| Duration of operation 90–150 minutes | Mulu W. et al./2013 [16] | 3 | 6 | 11 (-18.41–40.41) | 11 (-18.41–40.41) | - |
| Hospital stay >5 days | Mulu W. et al./2013 [16] | 2 | 3 | 8.2 (5.2–11.2) | 5.32 (0.01–10.65) | 89.8% |
| | Alemayehu T. et al./2019 [10] | 58 | 183 | 2.76 (1.13–4.37) | | |
| Age 1–14 years | Yallew WW. et al./2016 [8] | 14 | 148 | 0.25 (-0.06–0.56) | 0.25 (-0.06–0.56) | - |
| Admission to the surgery ward | Yallew WW. et al./2016 [8] | 75 | 240 | 2.86 (1.33–4.38) | 2.86 (1.33–4.38) | - |
| Patients admitted at Felege Hiwot Hospital | Yallew WW. et al./2016 [8] | 74 | 261 | 1.99 (1.2–2.77) | 1.99 (1.2–2.77) | - |
| Immuno-deficiency | Yallew WW. et al./2017 [29] | 31 | 92 | 2.34 (0.57–4.1) | 2.34 (0.57–4.1) | - |
| Central vascular catheter | Yallew WW. et al./2017 [29] | 5 | 4 | 6.92 (-11.17–25.01) | 6.92 (-11.17–25.01) | - |
| Patient received antimicrobial | Yallew WW. et al./2017 [29] | 104 | 294 | 8.63 (-1.79–19.05) | 8.63 (-1.79–19.05) | - |
| Medical waste container at room | Yallew WW. et al./2017 [29] | 102 | 431 | 0.18 (-0.290–0.65) | 0.18 (-0.290–0.65) | - |
| Previous hospitalization | Ali S. et al./2018 [29] | 20 | 25 | 1.65 (0.91–2.39) | 2.13 (0.71–3.55) | 49.4% |
| | Gebremeskel S. et al./2018 [31] | 27 | 43 | 3.22 (1.16–5.28) | | |
| Age 18–30 years | Ali S. et al./2018 [9] | 28 | 179 | 0.54 (0.22–0.85) | 0.54 (0.22–0.85) | - |
| Chest tube insertion | Ali S. et al./2018 [9] | 3 | 1 | 4.14 (1.57–6.71) | 4.14 (1.57–6.71) | - |
| Mechanical ventilation | Ali S. et al./2018 [9] | 12 | 22 | 1.99 (0.65–3.32) | 1.99 (0.65–3.32) | - |
| Malnutrition | Alemayehu T. et al./2019 [10] | 39 | 135 | 2.1 (0.78–3.41) | 2.1 (0.78–3.41) | - |
| Male sex | Gebremeskel S. et al./2018 [31] | 38 | 95 | 2.1 (0.45–3.67) | 2.1 (0.45–3.67) | - |
| Hospital stay < 5 days | Gebremeskel S. et al./2018 [31] | 5 | 119 | 0.03 (-0.01–0.07) | 0.03 (-0.01–0.07) | - |

**Note:** $I^2$: reported for the pooled effect of two or more studies.

University Hospital, Ethiopia was 207.6/1000 patient-days [9]. This highest incidence may be augmented by the severity of the disease [36] among ICU patients. Added, highly invasive procedures like intubation, peripheral, and central venous catheters are highly likely among ICU patients. Consequently, the risk of HCAI is higher among intubated patients and those on vascular catheterization [44].

In the time trend analysis, we found that HCAI was slightly increasing in Ethiopia from 1983 to 2017. The possible reasons might be more emphasis given on healthcare coverage than quality, increase in technological advancement, and overutilization of invasive procedures. Evidence revealed that advances in life-saving medical practices increase exposure to invasive procedures which increase the occurrences of nosocomial infections [11]. On top of this, nurses' burnout might contribute to the increasing trend. Evidence in the United States (US) revealed that nurses' burnout was found as a single most important associated factor for increased nosocomial urinary tract infection (UTI) and surgical site infection (SSI) [45].

The current systematic review and meta-analysis revealed surgical procedure and underlying non-communicable disease as determinants of HCAI. Accordingly, patients who had the surgical procedure were 3.37 times more likely to acquire HCAI as compared to patients who have no surgical procedure. The finding is in line with previous studies done in South Africa [7] and Poland [44]. The reason for the observed association could be explained by less

compliance to hand hygiene practice and high prevalence of surgical site infection (25.22%) in Ethiopia [46]. Compliance with hand hygiene practice is pivotal for the prevention and control of nosocomial infection, but only 3.6% keep hand hygiene before performing aseptic procedures at Debre Birhan referral hospital, central Ethiopia [47]. In 2017, the overall compliance to hand hygiene practice was 18.7% [48] and 22% [47] at Hiwot Fana Specialized Hospital and Debre Birhan Referral Hospital in Ethiopia, respectively. Consequently, unable to keep and maintain hand hygiene practice increased the acquisition of HCAI.

In this systematic review and meta-analysis, the odds of having HCAI among patients who have underlying non-communicable disease were nearly 3 times higher than their counterpart. This finding is supported by studies that reported positive association of HCAI with diabetes mellitus [49] and underlying renal disease [50]. The possible explanations for the observed association in the current study might be due to high prevalence of underlying diseases and the immune-suppressive effects of these diseases. In Ethiopia, a meta-analysis of studies showed high burden (6.5%) of diabetes mellitus [51]. Besides, another meta-analysis in other setting (North America, Europe, Latin America, and the Caribbean) reported immune-suppression as risk factor for HCAI [49]. Thus, higher odds of HCAI may be ascertained to the aforementioned antecedents.

## Strengths and limitations of the study

This systematic review and meta-analysis was the first national report on the prevalence of HCAI and its determinants in Ethiopia. However, it may lack national representativeness because no data were found from Benishangul Gumuz, Afar, Gambella, Somalia, Dire Dawa, and Harari regions of the country. Besides, the use of only English language, the absence of grey pieces of literature, and the unlimited time-period for the inclusion of studies may limit the conclusiveness of the finding. On top of this, the time-trend analysis might not reflect the exact trend because all the considered years didn't have reported data.

## Conclusions

The prevalence of healthcare-associated infection has remained a problem of public health importance in Ethiopia. Based on the subgroup analysis, the highest prevalence of HCAI was found in ICU followed by pediatrics, surgical, and obstetrics wards in descending order. Surgical procedures and underlying non-communicable disease were found as determinants of HCAI. Therefore, policy-makers and program officers should give due emphasis to the prevention of healthcare-associated infection with more attention for patients admitted to ICU. Furthermore, the existing infection prevention and control practices for patients who had surgical procedures and underlying non-communicable disease should be strengthened in Ethiopia.

## Supporting information

**S1 File. PRISMA checklist.**
(DOC)

## Author Contributions

**Data curation:** Demeke Mesfin Belay, Demewoz Kefale Mekonen, Biniam Minuye Birhan, Wubet Alebachew Bayih.

**Formal analysis:** Abebaw Yeshambel Alemu.

**Methodology:** Abebaw Yeshambel Alemu, Aklilu Endalamaw.

**Software:** Aklilu Endalamaw.

**Visualization:** Demeke Mesfin Belay, Demewoz Kefale Mekonen, Biniam Minuye Birhan, Wubet Alebachew Bayih.

**Writing – original draft:** Abebaw Yeshambel Alemu.

**Writing – review & editing:** Abebaw Yeshambel Alemu, Aklilu Endalamaw, Biniam Minuye Birhan, Wubet Alebachew Bayih.

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
