## [Decision Letter · Decision Letter 0]

24 Sep 2020

PONE-D-20-09587

Prevalence and risk factors of healthcare-associated infection in Ethiopia: a systematic review and meta-analysis

PLOS ONE

Dear Abebaw Alemu Yeshambel,

Thank you for submitting your manuscript to PLOS ONE. After careful consideration, we feel that it has merit but does not fully meet PLOS ONE’s publication criteria as it currently stands. Therefore, we invite you to submit a revised version of the manuscript that addresses the points raised during the review process.

We look forward to receiving your revised manuscript.

Kind regards,

Professor Kwasi Torpey, MD PhD MPH

Academic Editor

PLOS ONE

Journal Requirements:

2. At this time we ask that you include in your manuscript an explanation for the pooling of prevalence data from case-control studies and cross-sectional studies.

Additional Editor Comments (if provided):

The manuscript titled " Prevalence of risk factors of health associated infections in Ethiopia: a systematic review and meta-analysis" is an important addition in understanding the factors associated with health associated infections within the Ethiopian context. Though the manuscript makes an important contribution it is not publishable in its current form as a result numerous language errors. The manuscript should be reviewed by a native speaker and copyedited thoroughly.

1. Below are some examples (not exhaustive)

a. Abstract Background: emerging of multidrug resistance microbial infection............. emerging should read emergence

b. Following sentence on survey or surveillance should be revised for clarity

c sentence with varied instead of varying

d. Methods: Addis Ababa university: University must start with a capital letter

e. 14,240 were participated for prevalence estimates - Correct language error. participants?

f. Last paragraph on background : Varieties of studies can read as a number of studies

Study selection

g. Screening of title and abstract: This should be plural

h. Quality Assessment: Disagreement was resolved by interference by third reviewer. Appropriate word could be involvement, assessed etc

plus many more

2. a. Reference #9 : remove CAPS

b. #20 - Check

Reviewers' comments:

Reviewer's Responses to Questions

**Comments to the Author**

1. Is the manuscript technically sound, and do the data support the conclusions?

Reviewer #1: Yes

Reviewer #2: No

2. Has the statistical analysis been performed appropriately and rigorously? 

Reviewer #1: Yes

Reviewer #2: Yes

3. Have the authors made all data underlying the findings in their manuscript fully available?

Reviewer #1: Yes

Reviewer #2: Yes

4. Is the manuscript presented in an intelligible fashion and written in standard English?

Reviewer #1: No

Reviewer #2: No

5. Review Comments to the Author

Reviewer #1: Title : Prevalence and risk factors of healthcare-associated infection in Ethiopia: a systematic review and meta-analysis

General Comments

o It will be good express why the research focused only “patients because “Healthcare-associated infection” include acquired infection in the health care set up for patients, healthcare workers, and visitors.

o It would be good to see subgroup analysis and observe the prevalence of HCAI in each ward by observing the I2 , rather than “region, study design, diagnostic method and sample size” this may give good insight for the reader as well as specific decision actions?

Specific comments

Title:

Abstract:

1. in the abstract section “Addis Ababa university repository.” It is not a research data base but it is a student’s thesis data base, better to use as unpublished grey literature source rather than consider as a scientific repository site?

2. Is it possible to say “The national prevalence of healthcare-associated infection” for this pooled result because, the exposure of Healthcare workers and patients vary from type of facility and vary ward to ward? Did you get similar population to pool the prevalence of HCAI in Ethiopia, for example it may vary , ICU to ophthalmology and Teaching hospital and district hospital in Ethiopian condition?

Method:

o Is there any reason “There was no restriction of the study period” for this dynamic scientific world, don’t you consider the implementation of IP programs procedural and guidelines change have impact the prevalence of Health care associated infections in Ethiopia? For example, Infection guideline development since 2004 GC has an impact in Ethiopia to manage IP practices? How do you see this?

o “Not applicable because no primary data were collected. Did the protocol published? , why not ?

“

Result:

o Subgroup analysis section: it is good to conduct and observe subgroup analysis?

o For risk factors pooled value what was the measure of effect? Is that OR or RR, , don’t you think papers must be similar?

Discussion:

o Start the discussion with one summary paragraph.

o In the third paragraph of the discussion section “ The possible reasons for high prevalence in this study might be very low hand hygiene practice by physicians,” it needs a reference? May be should be supported by evidence?

Strength and limitation

Some of the limitation that I observed in this manuscript

- 1. Only one single language English for the inclusion criteria

- 2. Absence of Grey literature in the pooled result

- 3. Absence of time period for the inclusion criteria?

Reviewer #2: Prevalence and risk factors of healthcare-associated infection in Ethiopia: a systematic review and meta-analysis

This study is supposed to be an important work considering the escalation of healthcare-associated infections globally, however, the major drawback noticed is the lack of the systematic review of the identified studies. There is an obvious lack of information on healthcare-associated infections in the manuscript vis-a viz the different types and how the integration of these was done to culminate into meta-analysis.

Please, see comments in the reviewed submitted attachment.

All the best.

6. PLOS authors have the option to publish the peer review history of their article (what does this mean?). If published, this will include your full peer review and any attached files.

Reviewer #1: **Yes: **walelegn worku yallew

Reviewer #2: No

---

## [Author Response · Author response to Decision Letter 0]

6 Oct 2020

Point by a point response letter

Dear Academic Editor (Professor Kwasi Torpey, MD Ph.D. MPH), Reviewer #1, and Reviewer #2

After going through the entire manuscript, you forwarded your constructive comments which we missed to touch. Therefore, we are glad enough to express our sincerest thanks for your constructive editorial comments that could help to improve the novelty of our effort.

Editors general comments 

Comment: 1. Please ensure that your manuscript meets PLOS ONE's style requirements, including those for file naming. The PLOS ONE style templates can be found at

Author response: Yes indeed, accessing the PLOS ONE style templates from the given links, our manuscript has been made to meet PLOS ONE's style requirements, including those for file naming. These changes were made to meet PLOS ONE's style requirements and found throughout the revised version of the manuscript. 

Comment: 2. At this time we ask that you include in your manuscript an explanation for the pooling of prevalence data from case-control studies and cross-sectional studies.

Author response: Yes Sure! We use case-control and cross-sectional studies together to estimate the pooled effect of the determinants of healthcare-associated infection. But, we did not pool the prevalence from case-control and cross-sectional studies together unless specified. Nonetheless, the authors hadn’t encountered a prevalence report from a case-control study, as depicted in Table 1 the prevalence estimate of a pocket study was not presented under the prevalence column for a case-control study included. Now we include a statement that specifies the purpose of including case-control studies in the methods section under the sub-heading inclusion and exclusion criteria. 

Comment: 3. Please include captions for your Supporting Information files at the end of your manuscript, and update any in-text citations to match accordingly. Please see our Supporting Information guidelines for more information: http://journals.plos.org/plosone/s/supporting-information

Author response: Thank you. We include captions for supporting information files and update the in-text citations accordingly. You can find the caption at the end of the manuscript file following the figures caption. 

Additional Editor Comments (if provided):

Comment: The manuscript titled " Prevalence of risk factors of health-associated infections in Ethiopia: a systematic review and meta-analysis" is an important addition in understanding the factors associated with healthcare-associated infections within the Ethiopian context. The manuscript should be reviewed by a native speaker and copyedited thoroughly.

1. Below are some examples (not exhaustive)

a. Abstract Background: emerging of multidrug resistance microbial infection............. emerging should read emergence

b. Following sentence on survey or surveillance should be revised for clarity

c sentence with varied instead of varying

d. Methods: Addis Ababa university: University must start with a capital letter

e. 14,240 were participated for prevalence estimates - Correct language error. participants?

f. Last paragraph on background : Varieties of studies can read as a number of studies

Study selection

g. Screening of title and abstract: This should be plural

h. Quality Assessment: Disagreement was resolved by interference by third reviewer.

Appropriate word could be involvement, assessed etc

plus many more

2. a. Reference #9 : remove CAPS

b. #20 – Check

Authors’ response: Sure! We have tried our best to address the comments from the reviewers. Moreover, from repeated proof-reading of the manuscript, we found several grammatical errors, interlinings, police titles, punctuation errors, wordings, and spelling errors. Therefore, finding our colleague who has a Master of Arts in English, we have done our best to thoroughly copyedit the manuscript for English language usage. These editorial changes are found throughout the revised version manuscript.

Reviewers' general comments:

Reviewer's Responses to Questions

Comments to the Author

Comment: 1. Is the manuscript technically sound, and do the data support the conclusions?

Reviewer #1: Yes

Reviewer #2: No

Authors’ response: The design of our research is systematic review and meta-analysis. Based on the quality and strength level of information, evidence from systematic review and meta-analysis is the strongest. The finding is reported based on PRISMA guidelines and the sample size for the current study is high. ________________________________________

Comment: 2. Has the statistical analysis been performed appropriately and rigorously?

Reviewer #1: Yes

Reviewer #2: Yes

Authors’ response: Thank you for your appreciation! ________________________________________

Comment: 3. Have the authors made all data underlying the findings in their manuscript fully available?

Reviewer #1: Yes

Reviewer #2: Yes

Author response: Thank you indeed!

Comment: 4. Is the manuscript presented in an intelligible fashion and written in standard English?

Reviewer #1: No

Reviewer #2: No 

Authors’ response: Sure! We have tried our best to address the comments from both reviewers. Moreover, from repeated proof-reading of the manuscript, we found several grammatical errors, interlinings, police titles, punctuation errors, wordings, and spelling errors. Therefore, finding our colleague who has a Master of Arts in English, we have done our best to thoroughly copyedit the manuscript for English language usage. These editorial changes are found throughout the revised version manuscript.________________________________________

Review Comments to the Author

Reviewer #1: Comments 

General Comments

Comment: It will be good express why the research focused only “patients because “Healthcare-associated infection” include acquired infection in the health care set up for patients, healthcare workers, and visitors.

Authors’ response: Firstly, the meta-analysis is depending on the original studies report. Secondly, whoever he/she is, once they acquired infection in healthcare settings, he/she is considered and treated as a patient. 

Comment: It would be good to see subgroup analysis and observe the prevalence of HCAI in each ward by observing the I-square, rather than “region, study design, diagnostic method and sample size” this may give good insight for the reader as well as specific decision actions?

Authors’ response: Yes sure! As per your comment, we do a subgroup analysis based on ward type and include in the revised version of the manuscript, and we found important findings indicating the highest (25.8%) prevalence of HCAI in the intensive care unit (ICU) and pediatrics ward (24.16%). The revision is included in Figure 4 in the results section. You can appreciate the result in the track change version of the manuscript. 

Specific comments

Comment: 1. in the abstract section “Addis Ababa university repository.” It is not a research data base but it is a student’s thesis data base, better to use as unpublished grey literature source rather than consider as a scientific repository site?

Authors’ response: Thank you. In fact, Addis Ababa University’s online research repository is not considered as a database as PubMed and other databases. Unpublished works of its own students’ research are deposited to the University online repository. So, we correct the statement as per your comment you can find it as track change on the revised document in the abstract and methods section. 

Comment: 2. Is it possible to say “The national prevalence of healthcare-associated infection” for this pooled result because, the exposure of Healthcare workers and patients vary from type of facility and vary ward to ward? Did you get similar population to pool the prevalence of HCAI in Ethiopia, for example it may vary, ICU to ophthalmology and Teaching hospital and district hospital in Ethiopian condition?

Authors’ response: One of the reasons to do systematic review and meta-analysis is the absence of pooled evidence for a country. Different fragmented studies found in different areas could not represent the national level. However, pooling these primary studies could represent the national level. By considering its strength and limitation, the current meta-analysis could be reported as being national prevalence because no data and/or other meta-analysis studies at the national level were published before ours. Besides, as per your comment subgroup analysis is done by ward type and the result is incorporated in the results section under subheading subgroup analysis as Figure 4. 

Regarding variations from hospital to hospital, we have found the studies were done at referral and above level hospitals so we were unable to compare the varaiations across district and referral hospitals. 

Method:

Comment: Is there any reason “There was no restriction of the study period” for this dynamic scientific world, don’t you consider the implementation of IP programs procedural and guidelines change have impact the prevalence of Health care associated infections in Ethiopia? For example, Infection guideline development since 2004 GC has an impact in Ethiopia to manage IP practices? How do you see this?

Authors’ response: Usually, the year of publication could not be limited in case of systematic review and meta-analysis. It is known that, the variation of reports between studies conducted at different time periods and across geographical locations. There may be different updates throughout a different time period. There may be an impact due to the involvement of guidelines, policy change, human resources increment, or else. With the presence of such kinds, it is recommended to pool all the available studies to have pooled data. And, though it is not statistically significant the trend analysis showed an increase in the trend of HCAI in Ethiopia as incorporated in the original manuscript in Figure 6. 

Comment: “Not applicable because no primary data were collected. Did the protocol published? , why not ?

Author response: Thank you indeed! We include the above statement under ethics approval and consent to participate section of the manuscript in the methods section because no primary data is collected from patients. However, currently, the protocol is registered in the PROSPERO database with a registration number CRD4202016676, and available online:https://www.crd.york.ac.uk/prospero/display_record.php?ID=CRD42020166761

Result: 

Comment: Subgroup analysis section: it is good to conduct and observe subgroup analysis?

Author response: Thank you. Yes, Sure! As per your comment, we add the subgroup analysis based on the ward type. You can see Figure 4 in the revised version of the manuscript at the results section under the subheading subgroup analysis. 

Comment: For risk factors pooled value what was the measure of effect? Is that OR or RR, don’t you think papers must be similar?

Authors’ response: For this review, we found only 3 studies with cohort study design, and the other is done through a cross-sectional study design. Of these, only one study’s RR pooled with OR. We may combine anything that's trying to estimate the same thing if we have the estimate and standard error, using the inverse-variance method. The question is whether it makes sense to pool them. We may have RR and OR, then assuming the OR is a good approximation to the RR in our case. Thus, we theoretically combine them. Moreover, when the outcomes are rare, or in conducting a nested case-control study then these are approximately equal and can readily be combined. Some researchers decide these are similar enough to combine; others do not. The judgment of the meta-analyst in the context of the aims of the meta-analysis will be required to make such decisions on a case by case basis. OR, RR and HR are all measures of relative risks. Thus, combining OR & RR is sometimes acceptable

Discussion:

Comment: Start the discussion with one summary paragraph.

Authors’ response: Thank you and we do it now. The changes made could be appreciated in the first paragraph of the discussion section.

Comment: In the third paragraph of the discussion section “The possible reasons for high prevalence in this study might be very low hand hygiene practice by physicians,” it needs a reference? May be should be supported by evidence?

Authors’ response: It is now supported by evidence and updates as per your comment.

Strength and limitation

Comment: Some of the limitation that I observed in this manuscript

- 1. Only one single language English for the inclusion criteria

- 2. Absence of Grey literature in the pooled result

- 3. Absence of time period for the inclusion criteria?

Authors’ response: Thank you! With due respect, we consider these as limitations too. It can be depicted in the strength and limitations section of the revised manuscript in track changes. 

Reviewer #2: Comments 

Comment: This study is supposed to be an important work considering the escalation of healthcare-associated infections globally, however, the major drawback noticed is the lack of the systematic review of the identified studies. There is an obvious lack of information on healthcare-associated infections in the manuscript vis-a viz the different types and how the integration of these was done to culminate into meta-analysis.

Author response: Thank you. Now all the comments are revised as per the comment. 

Specific comments by reviewer #2 taken from the attachment 

Comment: Editorial comments throughout the document 

Author response: Yes Sure! From repeated proof-reading of the manuscript, we found several grammatical errors, interlinings, police titles, punctuation errors, wordings, and spelling errors. Therefore, finding our colleague who has a Master of Arts in English, we have done our best to thoroughly copyedit the manuscript for English language usage. These editorial changes are found throughout the revised version manuscript.

Background section

Comment: In the background section “Please endeavor to select references that are archived such as that from the World Health Organisation, US Centres for Disease Control and Prevention, etc. Not internet materials for HCAI”

Author response: Yes indeed! We have revised as per your comment, and we use the definitions used by CDC. The changes made could be seen in the introduction section.

Comment: “There is a published document by WHO. Please cite that in the stead. See number 10 citation”

Author response: Thank you indeed. The citation is revised as per the comment given. The revision made is available in the introduction section as track change. 

Results section 

Comment: The number of pocket studies included were 19 but written as 18

Author response: Thank you for raising this issue. Yes, the total number of studies included for meta-analysis was 19, but the total number of studies used to estimate the prevalence was 18. Nevertheless, the 19th study was case-control which we use it for determinant estimation. So, dear reviewer #2, the discrepancy in the number of studies was due to this scenario. Now, the number of articles included is revised accordingly. You can find the revisions made in the abstract section and the results section as track changes. 

Comment. In the results section at subheading risk factors of HCAI, you recommend the risk factors to be presented in the table. “This is better presented as a table where all denominators are known before statistical analysis done on it”

Author response: Yes sure! The revision is made according to your comment and included in Table 3 in the revised version of the manuscript. The changes can be depicted in the results section under sub-heading Determents of healthcare-associated infection.

---

## [Editor Report · Decision Letter 1]

8 Oct 2020

Healthcare -associated infection and its determinants in Ethiopia: A systematic review and meta-analysis

PONE-D-20-09587R1

Dear Mr Abebew Alemu,

We’re pleased to inform you that your manuscript has been judged scientifically suitable for publication and will be formally accepted for publication once it meets all outstanding technical requirements.

Kind regards,

Professor Kwasi Torpey, MD PhD MPH

Academic Editor

PLOS ONE

Additional Editor Comments (optional):

Comments addressed. There are still a few language corrections needed. Final copyediting will be helpful
---

## [Editor Report · Acceptance letter]

15 Oct 2020

PONE-D-20-09587R1 

Healthcare-associated infection and its determinants in Ethiopia: A systematic review and meta-analysis 

Dear Dr. Alemu:

I'm pleased to inform you that your manuscript has been deemed suitable for publication in PLOS ONE. Congratulations! Your manuscript is now with our production department. 

Kind regards, 

on behalf of

Professor Kwasi Torpey 

Academic Editor

PLOS ONE